# Analysis of residential satisfaction: An empirical evidence from neighbouring communities of Rohingya camps in Cox's Bazar, Bangladesh

Bangkim Biswas[1]*, Md. Nasif Ahsan[1], Bishawjit Mallick[2,3]

1 Economics Discipline, Khulna University, Khulna, Bangladesh, 2 Faculty of Environmental Sciences, Chair of Environmental Development and Risk Management, Technische Universität Dresden, Dresden, Germany, 3 Institute of Behavioral Science (IBS), University of Colorado, Boulder, Colorado, United States of America

⊗ These authors contributed equally to this work.
* bangkimbiswas@gmail.com

**Data Availability Statement:** All relevant data are within the manuscript and its Supporting information files.

## Abstract

This study aims to understand the level of residential satisfaction of the host communities' aftermath of the influx of Rohingya in Bangladesh. A total of 151 household heads were randomly interviewed from Ukhiya and Ramu Upazila of Cox's Bazar district, Bangladesh. A residential satisfaction index is developed with a total of twenty-two variables comprised of four components- social environment (SE), neighbourhood environment (NE), public services and facilities (PS&F), and dwelling units (DU). The coefficients of the components indicate that the PS&F, SE, and NE impact much on the overall residential satisfaction compare to the DU. The analysis demonstrates that the people who have tertiary level education, who is Muslim and whose work opportunities remain the same as before, are more satisfied, but older people are less satisfied than younger. Besides, the degradation of social harmony, livestock and agricultural land losses, and decreased wages were the significant causes of dissatisfaction. These findings may contribute to taking appropriate policies and programs for the host communities taken by the government and non-government organizations.

## 1. Introduction

Rohingya influx in Bangladesh is a challenging question in recent times. These vulnerable and disenfranchised communities of Myanmar put grossly enormous stress on the local livelihood, ecosystem, and essential services in the host communities, mainly in Teknaf and Ukhiya Upazilas of Cox's Bazar, Bangladesh [1]. Humanitarian Exchange reported that the Rohingya people adversely affect the host communities by booking the agricultural fields, "which were the main income sources of the poor people," pushing up the food price, creating threats for the local day labourers (by lowering the wages) and so on [2]. There was ample evidence that Rohingya refugees go outside their camps and work as labour that reduces the local work

**Funding:** The authors received no specific funding for this work.

**Competing interests:** The authors have declared that no competing interests exist.

opportunities for the local inhabitants [1]. Such a drastic increase in the population has also created pressure on the local economy, public services, and infrastructure [3]. More importantly, the education system has been negatively impacted because both students and teachers are hired to work on the refugee response [3]. Again, Rohingya camps' construction has exhausted more than 2,000 hectares of forest and croplands. Study shows that around 700 ton/day firewood has been collected and it causes the disappearance of the forest [3], thus affecting the livelihood conditions of the local communities and environment [4]. The host communities are usually facing problems in getting timely public services and facilities. Also, Riley et al. [5] found massive environmental damage in the camp area, including food, security, and safety. These all together adversely influence the residential satisfaction of the communities near the camps.

Here, the "residential satisfaction" refers to the personal feelings and awareness regarding the living place, i.e., home [6]. The concept of 'residential satisfaction' has been employed in a variety of transdisciplinary contexts ranging from planning and architecture to psychology and philosophy [7]. In general, residential satisfaction can be understood from two perspectives: (1) physical, which includes "consistent with the ingredient and services," and (2) social, which contains "bringing up the social networks" [8,9]. Most of the studies have been assimilated on both subjective and objective features to assess residential satisfaction [10]. Such attributes of residential satisfaction include individual perception, gratification, aspiration [7], and dissatisfaction that is strongly linked to an individual's psychological aspect [10]. The subjective attributes include the socio-demographic and individual features and housing quality [8,10]. Mridha and Moore [11] claimed that physical attributes are less likely to impact overall residential satisfaction than social attributes. Residential satisfaction, however, differs due to the variations of non-physical and physical features [12]. Moreover, residential satisfaction largely depends on the household's actual needs and aspirations of the current housing situation [13,14]. Thus, the level of residential satisfaction differs from the changes in household needs and aspirations, which may also be influenced by the social movement or deteriorating amenities in the living environment [15].

Such an understanding of residential satisfaction does not consider how an interruption of the determining factors contributes to the overall satisfaction changes. For example, increasing social conflict reduces residential satisfaction, so the factors that directly contribute to growing social conflicts also indirectly contribute to overall satisfaction. The massive Rohingya influx of 2017 is one such factor for the communities in Cox's Bazar of Bangladesh. Several studies claim that after Rohingya influx, communities' livelihood and environment quality near the Rohingya camps have been too distorted, affecting their residential satisfaction. However, to the best of author's knowledge, there is no empirical evidence on such claims regarding residential satisfaction at the individual household level. Therefore, the present study considers such understanding of residential satisfaction and explores how the Rohingya influx has affected the residential satisfaction of the host communities living nearby Rohingya camps. Also, it explains how the influx of Rohingya has created multiple complexities for living in the same place. However, this study evaluates the factors that impact residential satisfaction. It also assesses the demographic and socio-economic conditions and the perceptions of different aspects of livelihood, such as the dwelling units, social environment, neighbourhood environment, and public services and facilities. In doing so, both the socio-environmental and spatial attributes are considered. Explicitly, the proximity to Rohingya camps plays a vital role in deteriorating residential satisfaction, i.e., the study assumes people who live closer to the Rohingya camps have larger chances to lose their residential satisfaction compared to the people who live in distant communities. Therefore, this study takes both the neighbouring communities (within 2 km) and distant communities (more than 10 km away) from Rohingya camps. The

outcomes of this research will add new knowledge and be supportive and practical to take policies and programs.

The following sections are organized as section one describes the methods, including the study area information, data, and analytical procedures, section three present the results, and section four presents a brief discussion on the findings, including the relevance of the state-of-the-art, where section five concludes.

## 2. Methodology

### 2.1 Study area

Cox's Bazar, a district of the Chittagong division, is located near the Bay of Bengal, the world's largest delta. It consists of 8 Upazilas, 71 Unions, 177 Mauzas, 989 Villages, 4 Paurashvas, 39 Wards, and 169 Mahallas [16]. The eight Upazilas of Cox's Bazar district are Chakaria, Cox's Bazar Sadar, Kutubdia, Maheskhali, Pekua, Ramu, Tekhnaf, and Ukhiya [17]. Two Upazilas, namely Ukhiya (located at 21.2833˚N 92.1000˚E) and Ramu (located at 21.4583˚N 92.1000˚E) were selected as study sites in order to assess residential satisfaction. The Ukhiya Upazila consists of five Unions—Haldia Palong, Jalia Palong, Raja Palong, Ratna Palong, and Palong Khali, with a total area 64, 694 acres and a total population of 207,379 across 37,940 households, whereas the Upazila Ramu consists of eleven Unions—Chakmarkul, Fatekharkul, Garjania, Idgar, Joarianala, Kachhapia, Khuniapalong, Kauarkhop, Rashid Nagar, Rajarkul, Dakshin Mithachhari with total area 96,794 acres and a total population of 266,640 across 47,904 households [17].

This study considers the communities within 2 kilometres of the Rohingya camp area as the neighbouring communities. As neighbouring communities of the Rohingya, the villages Kutupalong from Raja Palong Union and Purba Balukhali from Palong Khali Union were selected for field study where a total of 716,150 people across 166,717 households, and 18,697 Rohingya across 3,709 households, live across camps area in Ukhiya Upazila respectively [18]. Again, the villages Jungle Dhoya Palong from Khuniapalong Union and Char Para (Caynda) from Daskmin Mithachhari Union, Ramu Upazila, were selected non-neighbouring communities because there is no presence of Rohingya camps (Fig 1). These two villages are 15.92 km and 29.43 km away from Kutupalong, the world's largest refugee camp area, respectively.

In both neighbouring and distant communities, people are involved in agricultural activities. They cultivate various products such as rice, pulses, potato, garlic, onion, ginger, betel nut, betel leaf, wheat, groundnut, sugarcane, tobacco, rubber, and different vegetables. Besides, a few people work as day labourers, drivers, and shopkeepers [16]. The average monthly income of the people is around 10,210 BDT of the Ukhiya Upazila (neighbouring communities) and 12,250 BDT of the Ramu Upazila (non-neighbouring communities), respectively [19]. Similar kind of facilities like health, education, security, and others are available in both communities. The literacy rate of the non- neighbouring communities is comparatively lower than in neighbouring communities [17]. The necessary household infrastructure is similar in types and manner, although less or more variation is observed (Table 1).

### 2.2 Sampling and data collection

There was no formal institutional review board approval for the empirical work, but this study is conducted as a master thesis under the Economics Discipline of Khulna University, Bangladesh. Both the individual household interview and focus group discussion tools were employed to collect the data. A total of 151 households were randomly selected for an interview during September-October 2019. Employing a semi-structured questionnaire, the interviewers collected information from the randomly selected participants with their consent form

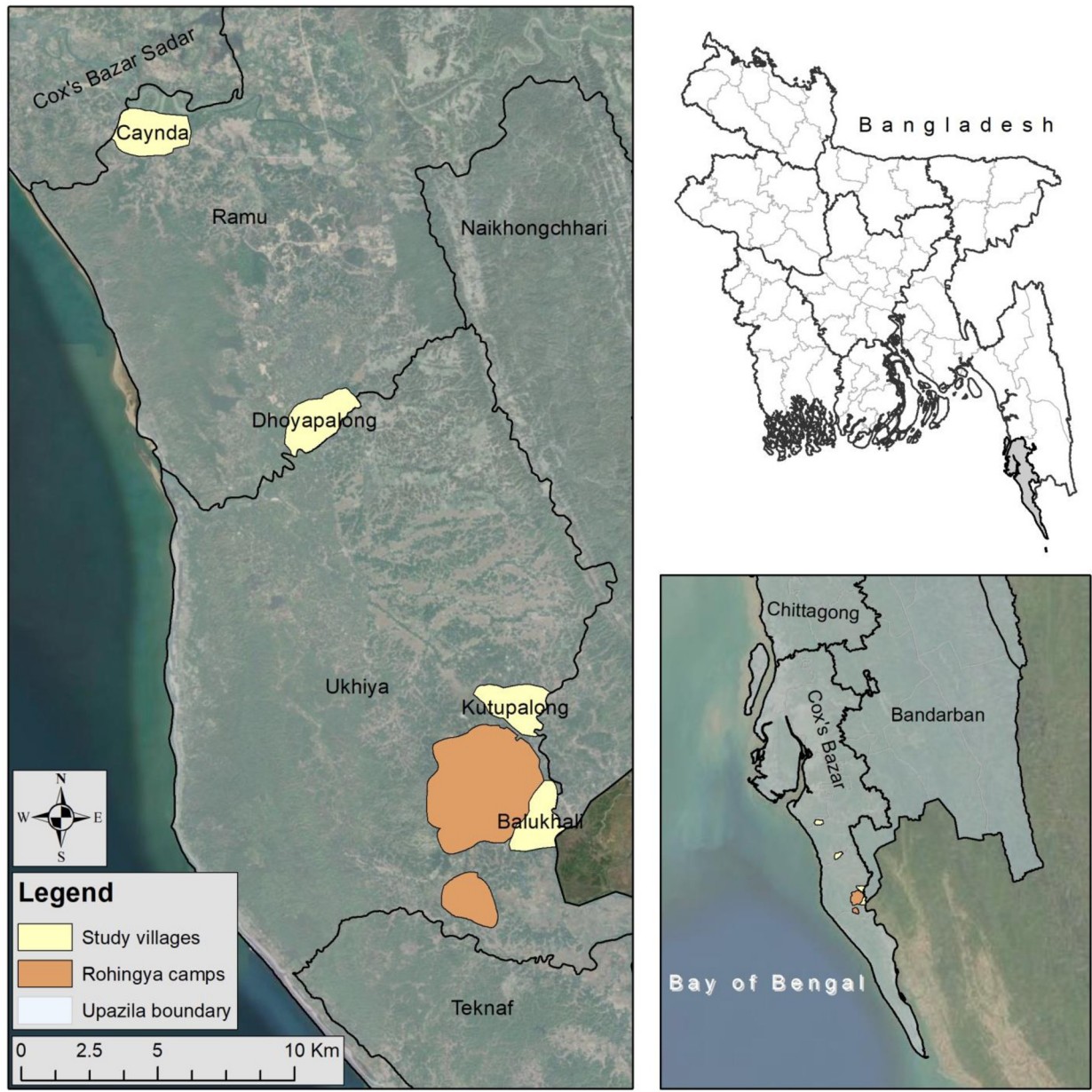

**Fig 1. Study area location.** Source: Authors' compilation, 2019.

four or five houses next in the selected study villages. The respondents were usually the household head and aged above 18 years. Before conducting the interview, the interviewers collected the oral and recorded consent of each participant in this empirical work.

The semi-structured questionnaire includes mainly three broad aspects of residential satisfaction related questions, namely: socio-economic, socio-demographic, and residential satisfaction. A pre-test of the survey questionnaire was performed. Necessary feedbacks from the pre-testing were addressed in the final version of the questionnaire. The final questionnaire was then structured a smart-phone assisted questionnaire development tool 'Kobotoolbox' (https://kootoolbox.org/), and data collection started. On average, every household interview took nearly 20–25 minutes to complete.

Table 1. Study villages, sample distribution, and basic household features.

| Attributes | | Ukhiya Upazila | | Ramu Upazila | |
|---|---|---|---|---|---|
| Union | | Raja Palong | Palong Khali | Khuniapalong | Daskmin Mithachhari |
| Villages | | Kutupalong | Purba Balukhali | Jungle Dhoya Palong (Dhoya Palong) | Char Para (Caynda) |
| Total household size | | 858 | 409 | 270 | 147 |
| Sample size | | 53 | 63 | 17 | 18 |
| Distance from Rohingya camp | | 0.85 | 0.42 | 15.92 | 29.44 |
| Literacy Rate (%) | | 36.1 | 53.3 | 23.3 | 31.3 |
| Housing Structure (%) | *Pucca* | 3.3 | 1.2 | 0.7 | 3.4 |
| | *Semi-Pucca* | 7.7 | 6.1 | 4.8 | 12.9 |
| | *Kutcha* | 66.7 | 84.1 | 78.9 | 81.6 |
| | *Jhupri* | 22.4 | 8.6 | 15.6 | 2.0 |
| Toilet Facility (%) | *Sanitary* | 31.7 | 45.9 | 27.2 | 26.5 |
| | *Non-sanitary* | 45.6 | 39.5 | 19.6 | 72.1 |
| | *None* | 22.7 | 14.7 | 53.0 | 1.4 |

Source: Authors' compilation base on BBS, 2011.

Besides, a total of six key informants' interviews (KII) and two focus group discussions (FGD) were conducted. School teachers and the members of local social clubs were the participants for the KIIs. One of the FGDs took place at a Hindu community, at Kutupalong, in which three women and five men participated and lasted around 30 minutes. Another FGD was conducted at village Balukhali, in which eight women and three men participated, and it lasted about 40 minutes. In the KIIs and FGDs session, different questions were asked to the participants principally, *(a) what problems were they facing in their current living place; (b) what were the social-bonding situation within the neighbourhoods after the Rohingya influx?; and (c) what do they think about their future mobility or future livelihood situation at their current living places?* Under these core questions, different issues were discussed during FGD sessions.

## 2.3 Analytical approaches

**2.3.1 Factors that measures residential satisfaction.** Residential satisfaction has a low affirmative relationship with neighbourhood facilities, but it is highly and positively correlated with the dwelling structures, social environment, support services, and public facilities [10]. Similarly, Hur and Morrow-Jones [20] stated that local government services and access to recreational opportunities are essential factors. Gan et al. [12] found three top factors, i.e., neighbourhood environment, affordability, and housing units, influence residence satisfaction. Again, Hur and Morrow-Jones [20] found that satisfaction with housing density and general appearances is a significant factor in the neighbourhood's satisfaction. The kinship and friends are also positively and significantly associated with residential satisfaction [21–23].

Mohit et al. [10] found that socio-demographic factors like age, household size, prior experience of residence, employment types, and working women influence residential satisfaction. Similarly, Tao, et al. [22] also observed that the household size is affirmatively related to residential satisfaction. Zanuzdana et al. [24], Lu [25], and Speare [26] observed that the socio-economic indicators (i.e., age, length of residence, etc.) are associated with residential satisfaction. Studies show that younger people with lower-income and education levels are less satisfied [22].

Interestingly, the length of stay and residential satisfaction are correlated [13,15]; however, it has both positive [27] and negative [28] influence on residential satisfaction. In particular, Dekker et al. [28] and Guillen-Royo et al. [29] claim that the elder members in the family and their health conditions (e.g., sickness) sometimes reduce the residential satisfaction. After all, the household economic conditions affirmatively impact residential satisfaction [28]. In contrast to these findings, Hur and Morrow-Jones [20] claim that income, education level, marital status, and race play no role in residential satisfaction.

Mohit et al. [10] claim that the housing infrastructure, like floor level and residency duration, is positively correlated with residential satisfaction. Similarly, Jiang et al. [14] found that housing dimensions, like the house, kitchen and bathroom facilities, technical and utility services, and proximity to the access road influence the overall residential satisfaction. In their study, Gan et al. [12] also claim that the dwelling unit includes housing size, bedroom(s), kitchen, bathroom, and common entrance are the essential features for housing satisfaction. However, very few dwelling factors significantly impact housing satisfaction; for example, the size positively affects residential satisfaction [30]. Table 2 represents significant indicators that influence residential satisfaction.

**2.3.2 Measurement of residential satisfaction.** The most common approach of quantifying and assessing the residential satisfaction is the self-evaluation either by asking the general question regarding the degree of satisfaction with residents' environment or asking the level of satisfaction with different aspects or components that, in some form, results in residential satisfaction index [32]. In line with the relevant indicators presented in Table 2 and discussed in section 2.3.1, this study considered four major components influencing residential satisfaction, namely: (1) social environment (SSE); (2) neighbourhood environment (SNE); (3) public services and facilities (SPSF); and (4) dwelling units (SDU). The list of the variables corresponding to the factors is presented in Table 2. Variables were assessed using a five-point Likert scale where 1 denoted extremely dissatisfied, and 5 denoted extremely satisfied. In creating the component, a reliability test between the corresponding variables was employed, and Cronbach's alpha score of 0.70 is considered for index building [33]. Results show that the internal consistency of the residential satisfaction components according to the Cronbach's alpha values of the SSE, SNE, SPSF, and SDU are 0.906, 0.920, 0.828, and 0.847, respectively.

This study follows the estimation technique employed by Mohit et al. [10] to create the residential satisfaction index, and it has two steps:

In step 1, the Satisfaction Component Index was determined for each of the four types of components at the household level using the following formula:

$$SI_c = \frac{\sum_{j=1}^{N} y_j}{\sum_{j=1}^{N} Y_j} \times 100 \tag{1}$$

Where,

SI$_c$ = Satisfaction value of component (c)

N = Number of variables of the component

y$_j$ = Score obtained by the household on the j$^{th}$ variables under the component

Y$_j$ = Maximum possible score on the j$^{th}$ variables under the particular component

In step 2, the overall residential satisfaction index score at the household level was determined using the following formula:

**Table 2. Key indicators of residential satisfaction.**

| Symbols | Variables | Key citations |
|---|---|---|
| $X_1$ | **Socio-demographic factors** | |
| $X_{11}$ | Age | [10,15,22,25,26] |
| $X_{12}$ | Gender | [20,22] |
| $X_{13}$ | Marital status | [20] |
| $X_{14}$ | Household size | [10,22] |
| $X_{15}$ | Race | [20,22] |
| $X_{16}$ | Distance from camp | [22] |
| $X_{17}$ | Length of the residence | [10,15,22] |
| $X_{18}$ | Number of children | [22] |
| $X_{19}$ | Years of education | [20,22] |
| $X_{110}$ | Number of sick persons | - |
| $X_{111}$ | Residential mobility | [22] |
| $X_2$ | **Socio-economic factors** | |
| $X_{21}$ | Employment type | [10] |
| $X_{22}$ | Working wives | [10] |
| $X_{23}$ | Family income | [13,15,20] |
| $X_{24}$ | Opportunity to work | - |
| $X_{25}$ | Property right | [15,22,23] |
| **(RSC) Residential satisfaction components** | | |
| SSE | **Social environment (SE)** | |
| $SE_1$ | Security from social crime | [23,31] |
| $SE_2$ | Cleanliness | [31] |
| $SE_3$ | Crowdedness | [31] |
| $SE_4$ | Social interaction/bonding | [23,31] |
| $SE_5$ | Pollution (water, air, sound) | - |
| $SE_6$ | Water supply | - |
| NE | **Neighbourhood environment (NE)** | |
| $NE_1$ | Garbage management | - |
| $NE_2$ | Density of housing | [20] |
| $NE_3$ | Greenery | [20] |
| PS&F | **Public services and facilities (PS&F)** [10,23] | |
| $SPF_1$ | Education facilities | [31] |
| $SPF_2$ | Health services | [1] |
| $SPF_3$ | Public transportation | [1] |
| $SPF_4$ | Access to recreational | [20] |
| $SPF_5$ | Relief intervention | - |
| $SPF_6$ | Union Parishad services | - |
| DU | **Dwelling unit (DU)** | |
| $DU_1$ | Housing size | [22,30] |
| $DU_2$ | Floor level | [10] |
| $DU_3$ | Kitchen | [10] |
| $DU_4$ | Dinning space/room | [10] |
| $DU_5$ | Bedroom | [10] |
| $DU_6$ | Toilet | [10] |
| $DU_7$ | Technical quality of dwelling | [14,22,23] |

Source: Authors' compilation based on literature review.

*Residential satisfaction index*:

$$SI_r = \frac{\sum_{j=1}^{N1} sdu_j + \sum_{j=1}^{N2} sne_j + \sum_{j=1}^{N3} sse_j + \sum_{j=1}^{N4} spsf_j}{\sum_{j=1}^{N1} SDU_j + \sum_{i=1}^{N2} SNE_j + \sum_{j=1}^{N3} SSE_j + \sum_{j=1}^{N4} SPSF_j} \times 100 \qquad (2)$$

where,

$SI_r$ = Overall residential satisfaction index score of a respondent which ranges from 20 to 90; $sdu_j$, $sne_j$, and $spsf_j$ represent the actual index score of an individual on the $j^{th}$ variables; $SDU_j$, $SNE_j$, $SSE_j$ and $SPSF_j$ represent the maximum possible score for the $j^{th}$ variables in the four components.

$N1 \ldots\ldots N4$ = Number of variables under the particular component of the residential environment.

**2.3.3 Exploring factors affecting residential satisfaction.** A multiple linear regression (MLR) model was applied to explore factors influencing overall residential satisfaction. The overall residential satisfaction index score is the dependent variable in the model. Four residential satisfaction components, SDU, SNE, SSE, and SPSF, including few socio-demographic and socio-economic variables, have been used as the independent variables (see Table 2). Since the outcome variable is continuous in manner, we ran like Mohit et al. [10] a multiple linear regression (MLR) model. The model is presented in Eq 3.

$$RS_i = \beta_0 + \beta_{1j}\sum_{j=1}^{10} X_{1ij} + \beta_{2j}\sum_{j=1}^{5} X_{2ij} + \beta_{3j}\sum_{j=1}^{4} RSC_{ij} + u_i \qquad (3)$$

Where,

*$i$ = 1, 2, 3,,,,,,,,,,,, n and j = 1, 2, 3,,,,,,,,,,,, m indicates the number of surveyed households and number of variables respectively.*

$RS_i$ denotes the residential satisfaction score of the $i^{th}$ respondent; $X_1$ denotes socio-demographic variables; $X_2$ denotes socio-economic variables; *RSC denotes* satisfaction components' scores; $\beta_1$, $\beta_2$ and $\beta_3$ are the parameters to be estimated; and $u_i$ denotes the error term.

To reduce the severity of the multicollinearity and explore the individual impact of residential satisfaction components, the MLR has been run five times. Before running these models, it was tested the cross-correlation among the independent variables. The first model includes the following variables: eight socio-demographic ($X_{1ij}$) variables: age ($X_{11}$), gender ($X_{12}$), household size ($X_{14}$), the distance of household from camp ($X_{16}$), length of the residence ($X_{17}$), the numbers of children ($X_{18}$), years of education ($X_{19}$), and numbers of sick persons ($X_{110}$) and three socio-economics variables: working wives ($X_{22}$), the opportunity to work ($X_{24}$) and property right ($X_{25}$). The second model includes the composite variable "social environments" ($SI_{SSEi}$), keeping other components outside the mode. It was assumed that people living close to the camp area are less satisfied in every residential component except the dwelling unit. To shrink the severity of co-linearity, the location variable ($X_{16}$) has been dropped. In the same way, incorporating a total of six socio-demographic, two socio-economic variables, and neighbourhood environment component ($SI_{SSEi}$) Model 3 has been run. Model 4 supposes that there is a correlation between the education level, employment types, and family income. The more educated people might find out jobs in the camp area, and they might earn more. Consequently, dropping the education and family income, Model 4 assessed the impact of employment type on residential satisfaction. Similarly, including seven socio-demographic, four socio-economics variables, and public services and facilities component ($SI_{SPS\&Fi}$), Model 4 has been run. Finally, dropping the years of education and employment variables, Model 5 has been run to see the actual effect of family income on residential satisfaction. Here, it was also

assumed that the dwelling units of both neighbouring and non-neighbouring communities are identical, and there is no co-linearity between the distance and dwelling units.

Finally, a Pearson correlation (r) test $[RS_i = f (SSE_{ij}, SNE_{ij}, SPS\&F_{ij}, SDU_{ij})]$ has been applied to examine all predictors' relationship and actual effect presented in Table 7 and constructed. "$H_0 = 0$: *there is no relationship between the variables*" as the null hypothesis and in contrast. "$H_A \neq 0$: *there is the relationship between the variables*" set up as the alternative hypothesis.

## 3. Results

### 3.1 Summary statistics of the surveyed households

The summary statistics presented in Table 3 show that the respondents' mean age is about 43 years (ranges between 19 to 75 years), and the average education level of them is around six years of schooling with a minimum of zero and a maximum of seventeen years of schooling presented. Amongst the respondents were 61% Muslim, 14% Hindu and 25% Buddha. And 88% of them were local and living in their birthplace. The summary statistics report that about 9% of the respondents have mobility experiences. Interestingly 15% of household heads are employed in a formal job like government or non-government sectors, whereas the rest of the household heads were involved in the different informal works such as day labourers, drivers, farmers, tailors, and barbers.

**Table 3. Summary statistics of surveyed households.**

| Variables | | Value |
|---|---|---|
| **Socio-demographic issues** | | |
| Age (Years)[a] | | 42.54 (±12.67) [19.00~75.00] |
| Gender (Male = 1, Otherwise = 0)[b] | | 70% |
| Marital status *(Married = 1, Otherwise = 0)* [b] | | 81% |
| Literacy level *(Years)* [a] | | 5.64 (±4.42) [0.00~17.00] |
| Household size *(Number)* [a] | | 5.13 (±1.81 [1.00~13.00] |
| Religion [b] | *Muslim* | 61% |
| | *Buddha* | 25% |
| | *Hindu* | 14% |
| Residential status *(By birth = 1, Otherwise = 0)* [b] | | 88% |
| Size of the floor *(Square feet)* [a] | | 513.05 (±982.56) [32.00~11250.00] |
| Distance from camp area *(Kilometer)* | | |
| | Neighbouring communities [a] | 0.62 (±0.46) [0.02~1.40] |
| | Non-neighbouring communities [a] | 22.87 (±6.89) [15.30~30.20] |
| Sick persons *(Number)* [a] | | 0.60 (±0.72) [0.00~3.00] |
| Mobility experiences *(Yes = 1, No = 0)* [b] | | 9% |
| **Socio-economic variables** | | |
| Employment type *(Formal = 1, Otherwise = 0)* [b] | | 15% |
| Working wives *(Yes = 1, No = 0)* [b] | | 15% |
| Property right *(Yes = 1, No = 0* [b] | | 94% |
| Monthly income *(BDT)* [a] | | 13536.42 (±11583.34) [2000.00~100000.00] |
| Monthly consumption *(BDT* [a] | | 10670.86 (±8032.45) [1800.00~70000.00] |

[a] Mean (standard deviation) [min~ max].

[b] Percentage.

Source: Authors' compilation 2019.

The average monthly income and consumption at the household level were 13,536 BDT (160 US$) and 10,670 BDT (125 US$), respectively, and demonstrates approximately 15% of household head's wives work for contributing their family earning. Although 87% of the total household heads live in their birthplace, 13% are migrants, and about 6% of them do not have property right they live in the Pube-land (The property of the Government that can use but don't have right to sell).

## 3.2 Ranking of the satisfaction

Total 22 residential satisfaction issues were considered and ranked based on the respondent's perception score. The results exhibited in Table 4 report that all seven issues related to dwelling units ranked in the first eight positions, for example, floor level (1st), floor size (3rd), bedroom (4th), dining space (5th), toilet (6th), kitchen (7th) and quality of dwelling (8th) have placed respectively.

Field observations report shows that most of the home is built on small hills, reducing inundation risk. The quality of transport systems was also not satisfactory for the residents living in the study villages. Furthermore, the respondents' were more dissatisfied with access to recreational facilities, greenery, pollution, crowdedness, housing density, safety from social crime, garbage management of the neighbourhood, and cleanliness (Table 4). The qualitative analyses also support these findings. For more explicit representation, these also have been displayed in Figs 2–5.

**Table 4. Ranking of the residential satisfaction issues of the neighbouring communities (N = 116).**

| Rank | Satisfaction with | V. S (5) | S (4) | I (3) | D (2) | V. D (1) | Total score |
|---|---|---|---|---|---|---|---|
| 1st | Floor level | 18.97 | 41.38 | 5.17 | 22.41 | 12.07 | 386 |
| 2nd | Community relationship | 11.21 | 26.72 | 34.48 | 19.83 | 7.76 | 364 |
| 3rd | Floor size | 9.48 | 41.38 | 1.72 | 32.76 | 14.66 | 346 |
| 4th | Bedroom | 2.59 | 45.6 | 7.76 | 31.90 | 12.07 | 342 |
| 5th | Dinning space | 5.17 | 44.83 | 2.57 | 30.17 | 17.24 | 337 |
| 6th | Toilet | 11.21 | 37.93 | 0.86 | 20.69 | 29.31 | 326 |
| 7th | Kitchen | 3.45 | 37.94 | 5.17 | 34.48 | 18.97 | 316 |
| 8th | Quality of dwelling | 5.17 | 37.07 | 1.72 | 31.90 | 24.14 | 310 |
| 9th | Relief intervention | 0.86 | 3.45 | 49.14 | 30.17 | 16.38 | 281 |
| 10th | Health services | 1.72 | 18.10 | 12.93 | 33.62 | 33.62 | 256 |
| 11th | Water supply | 0.86 | 6.90 | 20.69 | 31.03 | 40.52 | 228 |
| 12th | Union Parishad services | 3.45 | 6.03 | 15.52 | 32.76 | 42.24 | 227 |
| 13th | Education facilities | 1.72 | 11.21 | 8.62 | 30.17 | 48.28 | 218 |
| 14th | Cleanliness of the area | 1.72 | 3.45 | 12.93 | 35.34 | 46.55 | 207 |
| 15th | Garbage management | 0.00 | 2.59 | 9.48 | 50.00 | 37.93 | 205 |
| 16th | Safety from social crime | 0.86 | 8.62 | 9.48 | 18.97 | 62.07 | 194 |
| 17th | Density of housing | 0.00 | 1.72 | 8.62 | 42.24 | 47.41 | 191 |
| 18th | Crowdedness | 0.00 | 2.59 | 8.62 | 33.62 | 55.17 | 184 |
| 19th | Pollution | 0.00 | 0.00 | 7.76 | 38.79 | 53.45 | 179 |
| 20th | Greenery scenario | 0.00 | 0.00 | 5.17 | 32.77 | 62.07 | 166 |
| 21st | Access to recreational | 0.00 | 2.59 | 4.31 | 25.86 | 67.24 | 165 |
| 22st | Public transportation | 0.00 | 0.862 | 2.59 | 8.62 | 87.93 | 135 |

Note: V.S = Very satisfied, V = Satisfied, I = Indifferent, D = Dissatisfied, and V.D = Very dissatisfied.

Source: Authors' compilation, 2019.

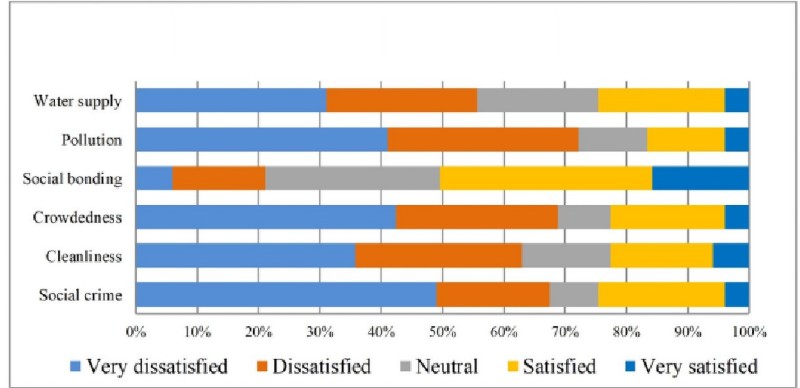

**Fig 2. Satisfaction with social environment.** Source: Authors' compilation, 2019.

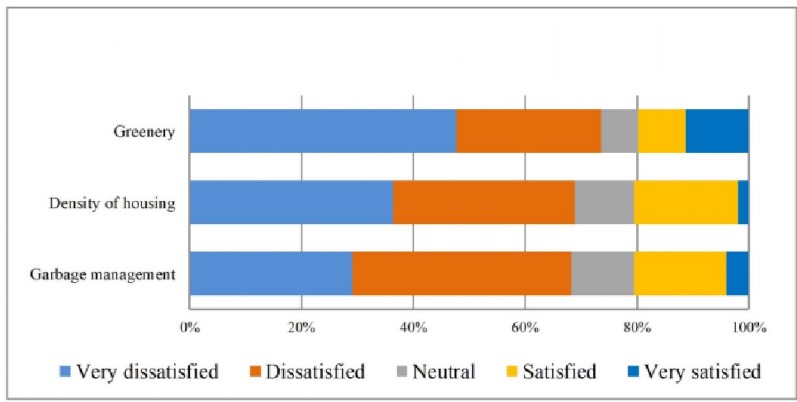

**Fig 3. Satisfaction with neighbourhood environment.** Source: Authors' compilation, 2019.

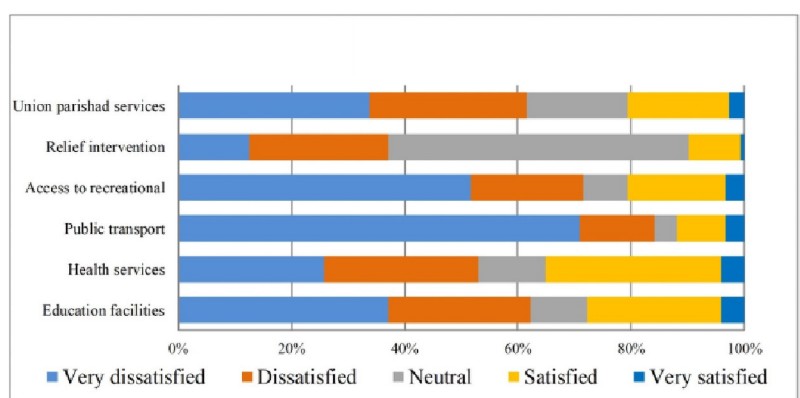

**Fig 4. Satisfaction with public services and facilities.** Source: Authors' compilation, 2019.

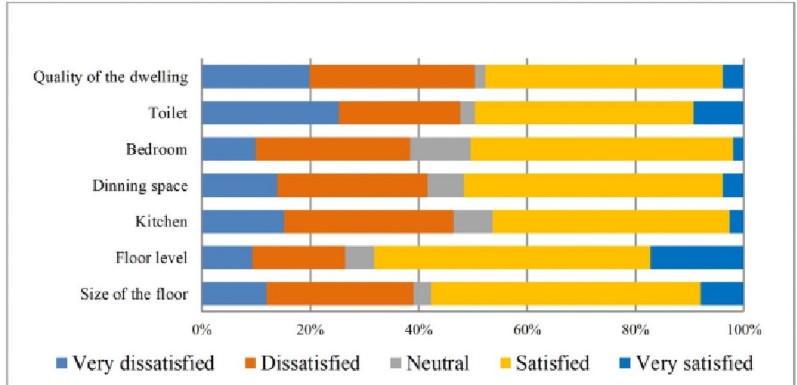

**Fig 5. Satisfaction with dwelling units.** Source: Authors' compilation, 2019.

### 3.3 Distribution of residential satisfaction scores

Fig 6 presents the scores of the four satisfaction components. The result shows that the studied households were more satisfied with dwelling units (60.66) compared to the social environment (48.17), public services and facilities (44.75), and neighbourhood environment (43.62) components. It indicates that the people were facing relatively more problems in their neighbourhood environment, public service and facilities, and social environment compared to the dwelling unit and being dissatisfied.

Similarly, Fig 7 presents the distribution of the overall residential satisfaction scores of the neighbouring and non-neighbouring communities. The bar diagram reports that the neighbouring communities' mean satisfaction score was around 43.60, with a minimum of 22.73 and a maximum of 62.73, respectively. In contrast to this, the non- neighbouring communities had a 73.77 mean satisfaction score, ranging between 63.64 and 85.45, respectively.

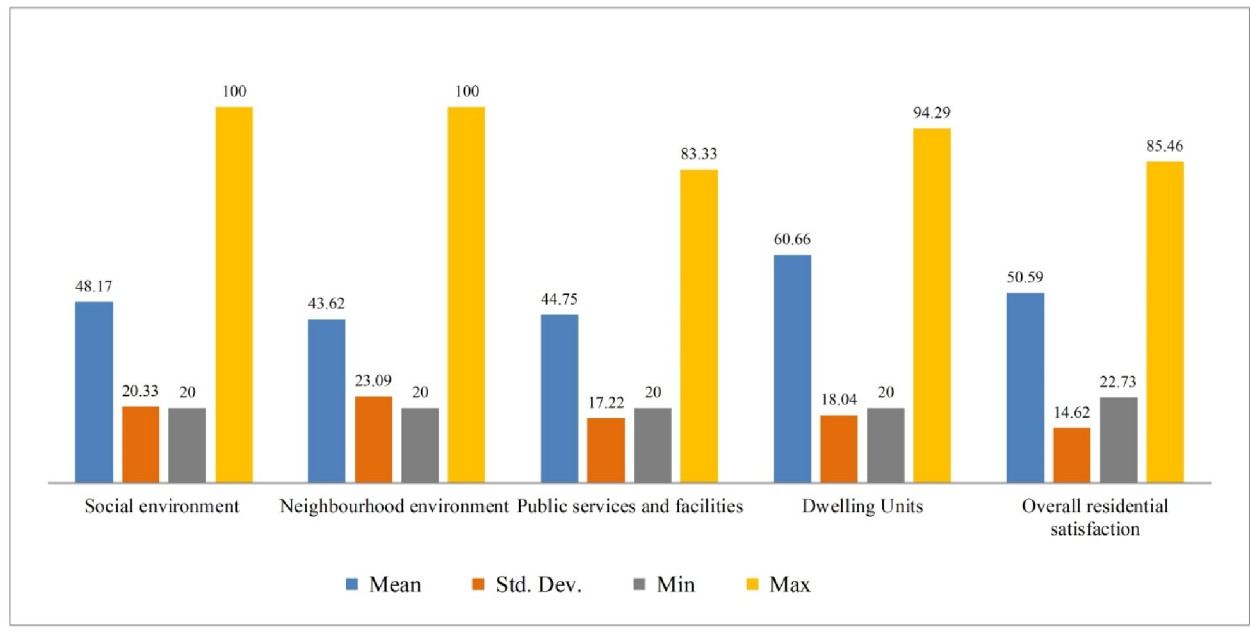

**Fig 6. Satisfaction score of four components.** Source: Authors' compilation, 2019.

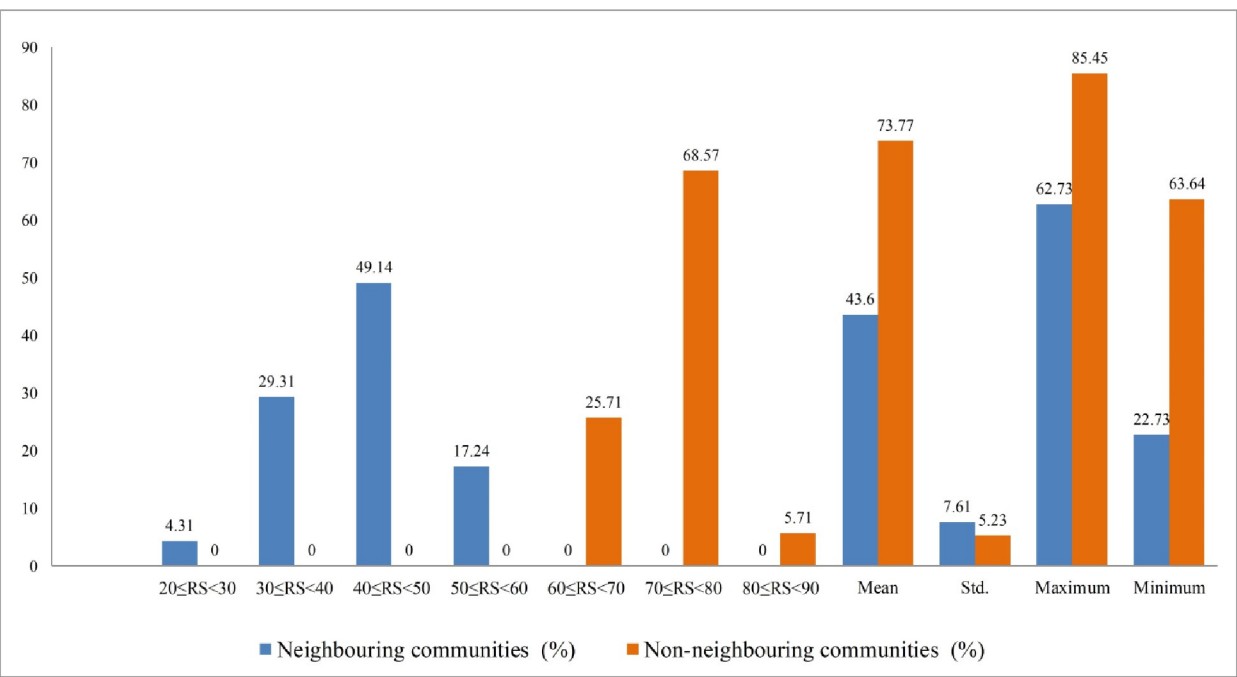

**Fig 7. Distribution of overall residential satisfaction.** Source: Authors' compilation, 2019.

It demonstrates that the neighbouring communities' satisfaction has been significantly differed with non-neighbouring communities due to the influx of Rohingya. The satisfaction score of the larger number of the neighbouring respondents (about 49.14%) and non-neighbouring respondents (about 68.57%) lies between 40≤RS<50 and 70≤RS<80, respectively. This implies that most neighbouring residents are less satisfied than non- neighbouring residents. The mean-scores of different residential components between the neighbouring and non- neighbouring are presented in Table 5. It suggests that the mean-scores of issues addressing residential satisfaction differ between the communities.

In particular, the neighbourhood environment and public services-and-facilities–issues revealed a higher degree of dissatisfaction in the neighbouring communities than their counterpart. All issues, except the toilet unit, addressing dwelling units were significantly different from zero, implying the neighbouring communities were less satisfied than non- neighbouring communities. The higher mean difference scores of social environment issues indicate that the neighbouring communities were more dissatisfied with that particular satisfaction issue.

### 3.4 Factors impact on residential satisfaction

The MLR regression model was run five times in order to reduce the severity of co-linearity and investigate the individual impact of residential satisfaction components. Table 6 represents the results of these five models. The first model reports that the household size is significantly and positively associated with residential satisfaction. The model shows that the older adults were much more dissatisfied, whereas people whose education level is *tertiary* were more satisfied than others. Also, the model estimated that people living nearer to the camp area were dissatisfied.

Model 2 suggests that household size and higher education levels were positively and significantly associated with overall satisfaction. It was because the higher educated persons had

**Table 5. Mean difference of residential satisfaction issues between neighbouring and non- neighbouring.**

| Satisfaction with | Neighbouring (N = 116) | Std. | Non- neighbouring (N = 35) | Std. | Mean diff. | t-value |
|---|---|---|---|---|---|---|
| **Social environment** | | | | | | |
| Security of social crime | 1.67 | 1.02 | 3.60 | 1.12 | -1.93 | -9.59*** |
| Cleanliness | 1.78 | 0.92 | 4.00 | 0.64 | -2.22 | -13.27*** |
| Crowdedness | 1.59 | 0.76 | 4.03 | 0.62 | -2.44 | -17.38*** |
| Social bonding | 3.14 | 1.10 | 4.23 | 0.60 | -1.09 | -5.60*** |
| Pollution | 1.54 | 0.64 | 3.83 | 0.79 | -2.29 | -17.57*** |
| Water supply | 1.97 | 0.99 | 3.91 | 0.66 | -1.95 | -10.96*** |
| **Index = SI$_{SSE}$** | ***38.97*** | ***12.05*** | ***78.67*** | ***9.26*** | ***-39.70*** | ***-17.94***** |
| **Neighbourhood environment** | | | | | | |
| Garbage management | 1.77 | 0.73 | 3.94 | 0.68 | -2.18 | -15.73*** |
| Density of housing | 1.65 | 0.71 | 3.91 | 0.51 | -2.27 | -17.50*** |
| Greenery | 1.43 | 0.59 | 4.31 | 0.80 | -2.88 | -23.18*** |
| **Index = SI$_{SSE}$** | ***32.30*** | ***10.50*** | ***81.14*** | ***9.63*** | ***-48.84*** | ***-24.57***** |
| **Public services and facilities** | | | | | | |
| Education facilities | 1.88 | 1.08 | 3.80 | 0.76 | -1.92 | -9.80*** |
| Health services | 2.21 | 1.15 | 3.91 | 0.66 | -1.71 | -8.39*** |
| Public transport | 1.16 | 0.49 | 3.06 | 1.35 | -1.89 | -12.65*** |
| Access to recreational | 1.42 | 0.70 | 3.94 | 0.59 | -2.52 | -19.32*** |
| Relief intervention | 2.42 | 0.84 | 3.23 | 0.55 | -0.81 | -5.36*** |
| Union Parishad services | 1.96 | 1.07 | 3.34 | 0.91 | -1.39 | -6.96*** |
| **Index = SI$_{SPS\&F}$** | ***36.84*** | ***9.42*** | ***70.95*** | ***9.41*** | ***-34.11*** | ***-18.79***** |
| **Dwelling Units** | | | | | | |
| Size of the floor | 2.98 | 1.31 | 3.69 | 0.80 | -0.70 | -3.00*** |
| Floor level | 3.33 | 1.34 | 4.06 | 0.42 | -0.73 | -3.18*** |
| Kitchen | 2.72 | 1.25 | 3.37 | 0.91 | -0.65 | -2.85*** |
| Dinning space | 2.91 | 1.29 | 3.31 | 0.90 | -0.41 | -1.76* |
| Bedroom | 2.95 | 1.17 | 3.34 | 0.87 | -0.39 | -1.84* |
| Toilet | 2.81 | 1.48 | 3.03 | 1.18 | -0.22 | -0.80 |
| Quality of the dwelling | 2.67 | 1.33 | 3.29 | 1.05 | -0.61 | -2.50** |
| **Index = SI$_{SDU}$** | ***58.20*** | ***18.56*** | ***68.82*** | ***13.48*** | ***-10.61*** | ***-3.14***** |

***Significant at 1% level,

** significant at 5% level and

* significant at 10% level.

Source: Authors' compilation, 2019.

found new jobs in the camps. Furthermore, the social environment component was found positive and significantly contributed to the overall residential satisfaction. The length of the residence exhibits a statistically significant inverse relationship with the satisfaction level of respondent households. It also predicts that people whose work opportunities have been reminded the same, their satisfaction level is higher.

Model 3 reveals that respondents having property rights were more satisfied than others who lack this right. Like Model 2 and 3 reports that education is positively, and the residence's length is negatively associated with residential satisfaction. It also predicts that the neighbourhood environment is positively and significantly affected the overall residential satisfaction. Model 4 estimates that the respondents who are Muslim and Buddha were more satisfied compared to Hindu. The coefficient of public support and facilities implies a positive association

**Table 6. Determinates of residential satisfaction (Index value).**

| VARIABLES | (1)<br>Model_1 | (2)<br>Model_2 | (3)<br>Model_3 | (4)<br>Model_4 | (5)<br>Model_5 |
|---|---|---|---|---|---|
| **Socio-demographic factors** | | | | | |
| *30≤ age<45* | -3.856* | 0.038 | -1.762 | -0.775 | -2.259 |
| | (2.097) | (2.032) | (2.018) | (1.973) | (1.594) |
| *45≤ age<60* | -4.963** | 0.291 | -2.343 | -0.654 | -2.178 |
| | (2.214) | (2.133) | (2.120) | (2.043) | (1.659) |
| *60≤ age75* | -4.558* | 0.0998 | -3.329 | -0.958 | -2.440 |
| | (2.503) | (2.459) | (2.410) | (2.357) | (1.885) |
| *age75+* | -9.573* | -1.847 | -3.728 | -7.849 | -6.485 |
| | (5.666) | (5.554) | (5.490) | (5.378) | (4.200) |
| Gender | 0.523 | 1.141 | 2.169 | 1.744 | 1.042 |
| | (1.423) | (1.379) | (1.358) | (1.361) | (1.160) |
| *Married* | | | | | 0.622 |
| | | | | | (1.445) |
| *Divorced* | | | | | 3.009 |
| | | | | | (3.272) |
| *Widow* | | | | | 1.981 |
| | | | | | (3.295) |
| Household size | 0.982** | 1.058** | 0.304 | 0.616 | 0.402 |
| | (0.475) | (0.464) | (0.459) | (0.452) | (0.391) |
| *Muslim* | | | | 3.335* | 5.719*** |
| | | | | (1.845) | (1.466) |
| *Buddha* | | | | 0.578 | 4.781*** |
| | | | | (1.957) | (1.612) |
| Distance from Rohingya camp | 1.154*** | | | | 1.040*** |
| | (0.0711) | | | | (0.0574) |
| Duration of residence | -1.645 | 2.893 | -3.337* | | -2.729* |
| | (1.901) | (1.872) | (1.837) | | (1.469) |
| No of children | -0.466 | -1.126** | -0.419 | -0.476 | -0.408 |
| | (0.583) | (0.565) | (0.564) | (0.563) | (0.456) |
| *High school* | -2.037 | -0.165 | 1.614 | | |
| | (1.465) | (1.444) | (1.408) | | |
| *College* | -2.408 | 6.768* | 3.402 | | |
| | (3.465) | (3.446) | (3.341) | | |
| *Tertiary level* | 10.97*** | 4.964* | 10.07*** | | |
| | (2.818) | (2.751) | (2.601) | | |
| Number of sick persons | -0.643 | -0.383 | -0.327 | -0.635 | -1.228 |
| | (1.059) | (1.032) | (1.013) | (0.979) | (0.799) |
| **Socio-economic factors** | | | | | |
| Employment type | | | | 2.422 | |
| | | | | (1.889) | |
| Working wife | -1.143 | 0.409 | 0.0158 | -0.354 | -0.123 |
| | (1.858) | (1.818) | (1.794) | (1.718) | (1.388) |
| *7000≤BDT<38000* | | | | | -1.572 |
| | | | | | (1.369) |
| *38000≤BDT<69000* | | | | | -2.891 |
| | | | | | (2.956) |

*(Continued)*

**Table 6.** (Continued)

| VARIABLES | (1) Model_1 | (2) Model_2 | (3) Model_3 | (4) Model_4 | (5) Model_5 |
|---|---|---|---|---|---|
| *69000≤BDT<100000* | | | | | -6.468 |
| | | | | | (6.475) |
| *Same* | 2.820 | 3.113* | | 2.829* | 1.662 |
| | (1.748) | (1.695) | | (1.622) | (1.316) |
| *Increased* | -1.300 | 1.430 | | -0.246 | -0.994 |
| | (1.993) | (1.935) | | (1.887) | (1.495) |
| Property right | 0.760 | 0.0673 | 5.751** | 2.223 | 1.565 |
| | (2.663) | (2.599) | (2.559) | (2.509) | (1.971) |
| **Residential satisfaction components (score value)** | | | | | |
| SSE | | 0.583*** | | | |
| | | (0.0346) | | | |
| SNE | | | 0.527*** | | |
| | | | (0.0272) | | |
| SPS&F | | | | 0.656*** | |
| | | | | (0.0414) | |
| SDU | | | | | 0.326*** |
| | | | | | (0.0284) |
| Constant | 44.21*** | 14.69*** | 23.79*** | 13.79*** | 22.37*** |
| | (3.975) | (4.270) | (3.958) | (3.907) | (3.562) |
| Observations | 151 | 151 | 151 | 151 | 151 |
| R-squared | 0.778 | 0.789 | 0.789 | 0.801 | 0.884 |

***Significant at 1% level,

** significant at 5% level and

* significant at 10% level.

Note: Age: 1 = age<29; Year of schooling: 1 = 0–5; Monthly income; 1 = BDT≤ 6999; Work opportunity: 0 = Decreased; Religion: 0 = Hindu and; Marital status:

0 = Unmarried are the reference categories.

Source: Authors' compilation, 2019.

with residential satisfaction. Finally, Model 5 exhibits similar results to Model 4, where the coefficients for both Muslim and Buddha were found statistically significant. Furthermore, there observed a statistically significant positive relationship between the dwelling unit component and the overall satisfaction.

### 3.5 The Pearson correlation test

The coefficient of Pearson correlation implies that the security of social crime (0.69), cleanliness (0.75), crowdedness (0.79), pollution (0.79), and water supply (0.71) are firmly correlated with overall residential satisfaction, whereas the social bonding (0.49) is moderately correlated at 1% level of significance (Table 7).

Results show that there is a significant relationship between social environment indicators and overall residential satisfaction. Similarly, the variables under the neighbourhood environment component show a strong correlation with the overall residential satisfaction.

Correspondingly, the satisfaction with access to recreational facilities (0.83) is extreme, education facilities (0.66), the public transportation system (0.70) and services from Union Parishad (0.60) are firmly, and health services (0.56) and relief intervention are (0.49) moderately correlated with the overall residential satisfaction. Finally, the test between the satisfaction

**Table 7. Pearson correlation test between residential satisfaction and different RSCs.**

| Explanatory variable | Mean | Std. | Pearson's (r) | P-value |
|---|---|---|---|---|
| **Satisfaction with social environment (SSE)** | | | | |
| Security of social crime | 2.12 | 1.32 | 0.69*** | 0.00 |
| Cleanliness | 2.30 | 1.27 | 0.75*** | 0.00 |
| Crowdedness | 2.15 | 1.26 | 0.79*** | 0.00 |
| Social bonding | 3.39 | 1.11 | 0.49*** | 0.00 |
| Pollution | 2.07 | 1.18 | 0.79*** | 0.00 |
| Water supply | 2.42 | 1.23 | 0.71*** | 0.00 |
| **Index = SI$_{SSE}$** | **48.18** | **20.33** | **0.86***** | **0.00** |
| **Satisfaction with the neighbourhood environment (SNE)** | | | | |
| Garbage management | 2.27 | 1.17 | 0.76*** | 0.00 |
| Density of housing | 2.17 | 1.17 | 0.82*** | 0.00 |
| Greenery | 2.10 | 1.38 | 0.82*** | 0.00 |
| **Index = SI$_{SNE}$** | **43.62** | **23.09** | **0.86***** | **0.00** |
| **Satisfaction with public services and facilities (SPS&F)** | | | | |
| Education facilities | 2.32 | 1.30 | 0.66*** | 0.00 |
| Health services | 2.60 | 1.28 | 0.56*** | 0.00 |
| Public transport | 1.60 | 1.11 | 0.70*** | 0.00 |
| Access to recreational | 2.01 | 1.26 | 0.83*** | 0.00 |
| Relief intervention | 2.61 | 0.85 | 0.49*** | 0.00 |
| Union Parishad services | 2.28 | 1.18 | 0.60*** | 0.00 |
| **Index = SI$_{SPS\&F}$** | **44.75** | **17.22** | **0.88***** | **0.00** |
| **Satisfaction with dwelling Unit (SDU)** | | | | |
| Size of the floor | 3.15 | 1.25 | 0.42*** | 0.00 |
| Floor level | 3.50 | 1.23 | 0.44*** | 0.00 |
| Kitchen | 2.87 | 1.21 | 0.43*** | 0.00 |
| Dinning space | 3.00 | 1.22 | 0.41*** | 0.00 |
| Bedroom | 3.04 | 1.12 | 0.42*** | 0.00 |
| Toilet | 2.86 | 1.41 | 0.21*** | 0.01 |
| Quality of the dwelling | 2.81 | 1.29 | 0.40*** | 0.00 |
| **Index = SI$_{SDU}$** | **60.66** | **18.04** | **0.53***** | **0.00** |

***Significant at.01 level,

** significant at.05 level and

* significant at.10 level.

Source: Authors' compilation, 2019.

with the dwelling unit and overall satisfaction level shows that all variables related to dwelling units significantly impact the residential satisfaction level, and almost all units have a moderate positive correlation. Grossly, the coefficient values of residential satisfaction components: public services and facilities (0.88), social environment (0.86), and neighbourhood environment (0.86) are more strongly related to the overall residential satisfaction level than the dwelling units (0.53).

## 3.6 How do the host communities perceive their satisfaction?

It is essential to know how the host communities handled complex problems after the influx of Rohingya. Fig 8 presents the findings observed from KIIs, FGDs, and semi-structural questionnaire. Firstly, a lot of forest area was destroyed to construct the Rohingya camps, besides the

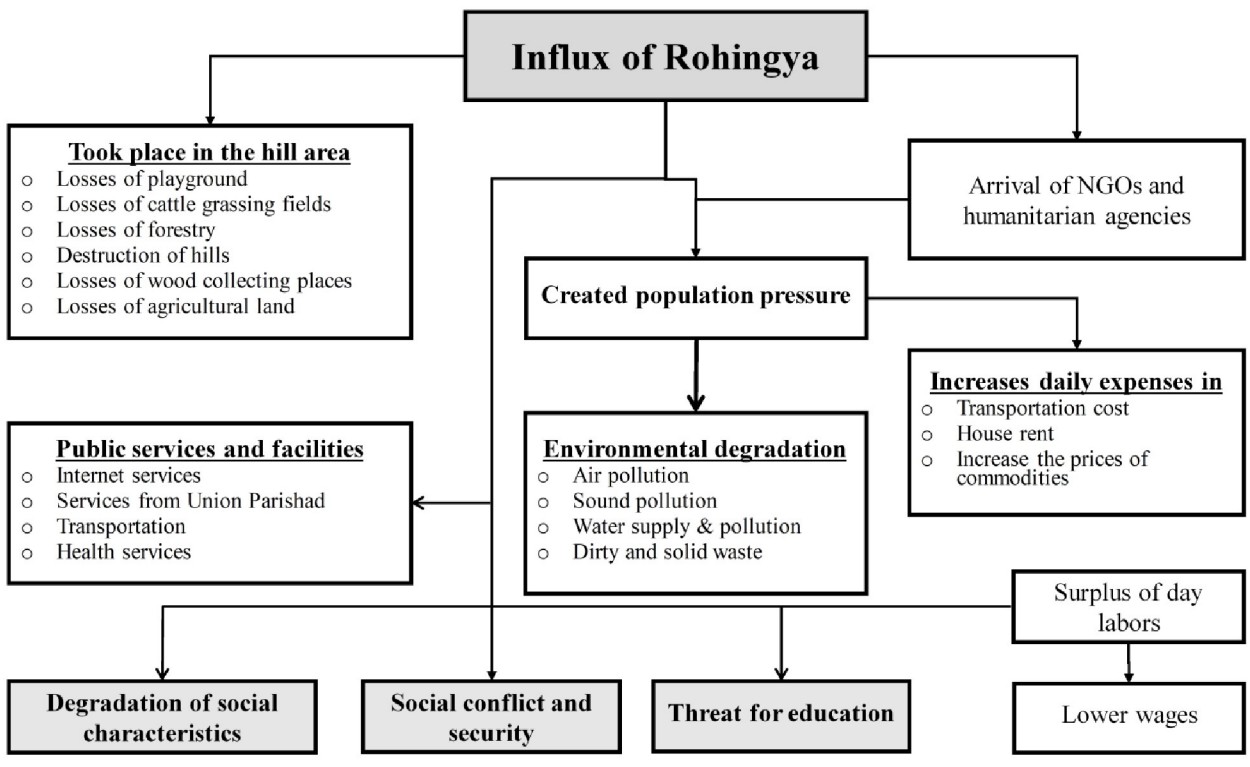

**Fig 8. Problems and dissatisfaction flow due to Rohingya influx.** Source: Authors' compilation, 2019.

host communities lost their playground, grazing fields, greenery, and wood collecting places. Different humanitarian agencies gathered there and created the local population pressure more, which negatively impacted the environment, such as air, sound, and water pollution. Furthermore, the increased population created a traffic jam that harmed passengers' and school-going children's time. Due to a large population, the daily expenses such as transportation cost, house rent, and price of necessary daily food were increased.

They face so much complexity to get the birth certificate, NID, required to move anywhere recently. Furthermore, the local communities informed that although there are separate clinics and hospitals for the Rohingya, they take services from the regular health clinics and hospitals that hamper the quality of services. Besides, there are severe problems with internet service and services from Union Parishad. Correspondingly, some local people claimed that it has been degrading the social characters' day by day and told that the youngers go there (camps) and take drugs and enjoy with young women and girls and even some young people got married to the Rohingya girls divorcing their wives. At a KII session in the BaluKhali Latifur Nessa Govt. primary school teachers claimed that the students' enrolment decreased. Students are learning swearwords. The guardians of the school-going children were afraid of their children. Even the non-registered institutions' teachers had given up the teacher's occupations and joined NGOs.

On the eve of the influx of Rohingya, although a few local educated persons have found the work opportunities, because of the surplus-labour, most of the day labourers have been lost their livelihood sources and being less-paid. However, comprehensively, these affect the social environment's satisfaction factors, neighbourhood environment, and public services and facilities.

## 4. Discussion

The unexpected influx of Rohingya plays a significant role in the residential satisfaction of the Bangladeshi people, who are living nearer to the camp areas. This study assesses the residential satisfaction and relevant aspects of four host communities. Findings show that the mean satisfaction score of the neighbouring and non- neighbouring communities is 43.60 and 73.77, respectively, which demonstrates that the influx of Rohingya has influenced the neighbourhood's overall residential satisfaction level. The satisfaction ranking results reveal that the neighbouring communities are less satisfied with regards to the social environment, neighbourhood environment, and public services and facilities than the dwelling unit component. Significantly, the ranking of satisfaction shows that people living in this area are very much dissatisfied with the public transportation system, access to recreational, greenery scenario, pollution, crowdedness, housing density, safety from social crime, social bonding, health services, educational facilities, services form Union Parishad and others.

The analysis shows that people with higher levels of education were more satisfied, where Hur and Morrow-Jones [20], in their study, stated that there is no role of education on satisfaction. Akin to Tao, et al. [22], this study found that the household size is positively related to the residential satisfaction level, where Mohit et al. [10] and Guillen-Royo et al. [29] reported that household size negatively impacts residential satisfaction level. The analysis shows that older people are more satisfied, supporting the prior studies conducted by Zanuzdana et al. [24] and Speare [26]. Although Mohit et al. [10] and Fang [27] predicted the duration of the residency is positively related to the residential satisfaction, in our study, we found that opposite association which is akin to the findings of Dekker et al. [28]. The result estimates that the people whose work opportunities remain the same as before are relatively much more satisfied than people whose work opportunities have been declined and increased. The satisfaction varies across the religion, such as the Muslim and Buddha are more satisfied than the Hindu. Also, the households that belong to more children and living near the camp area were less satisfied, but the people who have property rights are satisfied. The coefficients of residential satisfaction components demonstrate that the public services and facilities (0.66), social environment (0.58), neighbourhood environment (0.53) significantly impact much on the overall residential satisfaction compare to the dwelling units (0.33). The correlation coefficient of the Pearson estimation supports these findings. This study supports the finding of Hur and Morrow-Jones [20], who predicted the local government services and facilities play an essential role in increasing residential satisfaction. In their assessment of residential satisfaction, Mohit et al. [10] found a similar association, but they found that the dwelling structures, social environment, public service facilities are highly and positively related to residential satisfaction compare to the neighbourhood facilities.

## 5. Conclusion

This study assesses how and to what extent the Rohingya influx affects the host communities' residential satisfaction and investigates the key factors that impact overall residential satisfaction. This assessment's findings reveal a significant difference in the residential satisfaction score between the neighbouring and non-neighbouring communities. Remarkably, the people living near the camp areas are less satisfied with the transportation system, pollution, housing density, safety from social unrest, and criminal activities. However, findings suggest that the host communities' satisfaction level can be enriched by improving the public services and facilities like public transport system, education facilities, health services, access to recreational, relief intervention, and Union Parishad services. The improved quality of neighbourhood environment such as garbage management system, housing density, greenery, proper garbage

management system, building a house in the planned way, and restriction on haphazard defor-estation might increase the host communities' satisfaction level. Furthermore, it is essential to pay attention to the prohibition of child labour and school enrolment.

Notably, living in the same place together, the co-existence and resilience among both com-munities might be enhanced that interns might increase the host communities' satisfaction level, which can also be assessed in further research. However, this assessment of the host com-munities' residential satisfaction due to the influx added new values to the scientific discourses. The findings would support the government, non-government, and humanitarian agencies to take the host communities' appropriate policies and programs.

## Supporting information

**S1 Table. Descriptive statistics of the variables.**
(DOCX)

**S1 Dataset. Residential satisfaction dataset.**
(XLS)

## Acknowledgments

The authors are grateful to Babul Shaikh (Shraban) for his hard work during the field survey and Md. Yahya Tamim for his support in the map preparation of the study sites.

## Author Contributions

**Conceptualization:** Bangkim Biswas, Md. Nasif Ahsan, Bishawjit Mallick.

**Data curation:** Bangkim Biswas, Md. Nasif Ahsan, Bishawjit Mallick.

**Formal analysis:** Bangkim Biswas, Md. Nasif Ahsan, Bishawjit Mallick.

**Investigation:** Md. Nasif Ahsan, Bishawjit Mallick.

**Methodology:** Bangkim Biswas, Md. Nasif Ahsan, Bishawjit Mallick.

**Software:** Bangkim Biswas, Md. Nasif Ahsan, Bishawjit Mallick.

**Supervision:** Md. Nasif Ahsan, Bishawjit Mallick.

**Validation:** Md. Nasif Ahsan.

**Visualization:** Bangkim Biswas, Md. Nasif Ahsan, Bishawjit Mallick.

**Writing – original draft:** Bangkim Biswas, Md. Nasif Ahsan, Bishawjit Mallick.

**Writing – review & editing:** Bangkim Biswas, Md. Nasif Ahsan, Bishawjit Mallick.

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
