## [Decision Letter · Decision Letter 0]

16 Feb 2021

PONE-D-21-02960

Analysis of Residential Satisfaction: An empirical evidence from neighbouring communities of Rohingya camps in Cox's Bazar, Bangladesh

PLOS ONE

Dear Authors,

Thank you for submitting your manuscript to PLOS ONE. After careful consideration, we feel that it has merit but does not fully meet PLOS ONE’s publication criteria as it currently stands. Therefore, we invite you to submit a revised version of the manuscript that addresses the points raised during the review process.

We look forward to receiving your revised manuscript.

Kind regards,

Professor Hafiz T.A. Khan, Ph.D, CStat

Academic Editor

PLOS ONE

Journal Requirements:

5. We note that Figure 1 in your submission contain map images which may be copyrighted. All PLOS content is published under the Creative Commons Attribution License (CC BY 4.0), which means that the manuscript, images, and Supporting Information files will be freely available online, and any third party is permitted to access, download, copy, distribute, and use these materials in any way, even commercially, with proper attribution. For these reasons, we cannot publish previously copyrighted maps or satellite images created using proprietary data, such as Google software (Google Maps, Street View, and Earth). For more information, see our copyright guidelines: http://journals.plos.org/plosone/s/licenses-and-copyright.

5.1.    You may seek permission from the original copyright holder of Figure 1 to publish the content specifically under the CC BY 4.0 license. 

5.2.    If you are unable to obtain permission from the original copyright holder to publish these figures under the CC BY 4.0 license or if the copyright holder’s requirements are incompatible with the CC BY 4.0 license, please either i) remove the figure or ii) supply a replacement figure that complies with the CC BY 4.0 license. Please check copyright information on all replacement figures and update the figure caption with source information. If applicable, please specify in the figure caption text when a figure is similar but not identical to the original image and is therefore for illustrative purposes only.

Reviewers' comments:

Reviewer's Responses to Questions

**Comments to the Author**

1. Is the manuscript technically sound, and do the data support the conclusions?

Reviewer #1: Yes

Reviewer #2: Partly

Reviewer #3: Yes

2. Has the statistical analysis been performed appropriately and rigorously? 

Reviewer #1: Yes

Reviewer #2: Yes

Reviewer #3: No

3. Have the authors made all data underlying the findings in their manuscript fully available?

Reviewer #1: Yes

Reviewer #2: Yes

Reviewer #3: No

4. Is the manuscript presented in an intelligible fashion and written in standard English?

Reviewer #1: Yes

Reviewer #2: Yes

Reviewer #3: No

5. Review Comments to the Author

Reviewer #1: The paper has done an important empirical investigation on the satisfaction and perspectives of the communities who hosted over a million of Rohingyas in Bangladesh. It has huge policy implications. The mixed method of this study has helped design a robust study leading to aligned findings. The Sampling and data collection technique for this research seem well deigned. This study added new insight into the residential satisfaction research in the new context and assessed some new indicators of residential satisfaction like Relief intervention, services from local govt. and others. Since there is no formal institutional approval of the study it is important to understand how the security of the data is maintained. Further details about the both study areas regarding to the housing status, average monthly income and different govt. and other facilities will improve the paper. Also, the discussion on the policy implications can be further strengthened. Thorough editing of the paper is required to make it a standard paper for a reputed journal like PLOS ONE.

Reviewer #2: • The author reported an important topic on going policy challenges; there are very few scientific evidence exist for researching on this challenging and highly sensitive issue by looking on transboundary migrations, social conflicts.

• The study objective is “how and to what extent the host communities' residential satisfaction level has been influenced due to Rohingya”. The word “Influence” needs to be revised. Looking on the results it has reported more on individual’s perception on residential satisfaction; but drawing an robust to Rohingya camp remain open (given that it often hard to make a clear boundary in presence of many other factors).

• Include an operation definition of “neighboring”; in your sample case study unit they vary between 1 to 22 KM….

• Mention some scopes of the study

• Please highlight some criteria for survey unit (case study) selection – spatial and non-spatial; and mention about why the candidate unites were selected what limitation or sensitivity it may cause in the results.

• The author included almost a comprehensive methodology; however, it might be helpful to link and discuss on Multicriteria approach as they are dealing with multivariate factors; therefore it is possible to find some cluster in the in the response variables and fine tune possibility of the model fit. See some example:

http://www.macrothink.org/journal/index.php/emsd/article/view/4874

https://www.sciencedirect.com/science/article/abs/pii/S2212420915301023

• In general, the analytical approach needs a revisit by highlighting the objective of the research. It has built some type of index on RS and tried to investigate to key factors.

• In results, the fitting the linear regression curve is not much logical, where you the observation point are shows some clusters…it is obvious that RSI and SSE/SNE/RSI is correlated, the level might be highly due to the number of variables. Please check their sensitivity

• The word “comparative” should be used with care. So far, the objective and results of the research is exploring and reporting the KI opinion or perceptions. It is hardly compare the situation after and before…where there is less firm-base to compare, rather than explore on the individual perception on current state.

• The study has documented relevant literature, method and findings well; however still need some more re-look in discussion part. For instance, Need to generalize and link to theories and literature that has already been discussed earlier so that the scientific contribution become clear

• The author may consider to include a conclusion by highlighting the key findings, limitations and way forward

• There are some typo and minor errors which need to be revised as well

• Considering all points raised, this could be one of the important contributions for multidisciplinary readers of Plos Journal - those are interested on the study regional and beyond.

good luck...

Reviewer #3: This study aims to measure the level of residential satisfaction of the host community of Rohinga refugees, which is weakly addressed in this paper. Apart from the used tools to index the satisfaction level, the methodology is week. It would be necessary to know the level of satisfaction before the influx of the migrants so that the level of satisfaction can be compared.

1. The sampling technique is not straightforward- it is just not enough to say that the HH was randomly selected. It is okay even if the sample is not randomly selected if backed by valid motivation. In sampling, there is no indication of, discussion of the Non-Ukhiya group. How do we know if Ukhiya and Non-Ukhiya groups are comparable? We have to know if these groups are comparable by education, income, and other basic household characteristics.

2. Figure 7 compared the satisfaction level of Non-Ukhiya with the Ukhiya community and tried to establish that the cause of the difference is due to the influx of migrants (line 269). This is a very weak approach to establish this causality because we do not know the level of satisfaction of Ukhiya and Non-Ukhiya community before the influx of migrants. We only see the difference after the influx. The direction of difference can go to many ways. The difference may exist even before the influx of the migrants (outcomes of table 4 almost confirms the difference is pre-influx).

3. Why are 5 different models used to evaluate the level of satisfaction is not clear (precisely- each model is OLS with a different combination of socio-demographic variables)? What is the motivation of each model?

4. L155- I wonder if authors used a Likert scale, why do they need RSI ? There is a need for a detailed literature review backing this index's use.

5. As this study is about HH satisfaction analysis, OLS may not be the better approach, I would recommend authors to play with other commonly used satisfaction analysis methods such as Factors Analysis and Principal Components Analysis. These will enable identifying laten variables asoociated with satisfaction and preventing multicollinearity effects n the model estimations.

6. PLOS authors have the option to publish the peer review history of their article (what does this mean?). If published, this will include your full peer review and any attached files.

Reviewer #1: No

Reviewer #2: **Yes: **Sujit Kumar Sikder

Reviewer #3: No

---

## [Author Response · Author response to Decision Letter 0]

12 Apr 2021

Review of the manuscript

Manuscript number: PONE-D-21-02960

Title: Analysis of residential satisfaction: An empirical evidence from neighbouring communities of Rohingya camps in Cox's Bazar, Bangladesh 

Dear Editor, 

We are very obliged to all reviewers for their essential and thoughtful suggestions to improve our manuscript. In the revised manuscript, we tried to address all those valuable comments. The detailed responses to the reviewers’ questions are presented below:

Reviewer #1: 

The paper has done an important empirical investigation on the satisfaction and perspectives of the communities who hosted over a million of Rohingyas in Bangladesh. It has huge policy implications. 

- Thanks for your remarks, and we also think our findings will help to improve the future development initiatives for the study sites. 

The mixed method of this study has helped design a robust study leading to aligned findings. The Sampling and data collection technique for this research seem well deigned. This study added new insight into the residential satisfaction research in the new context and assessed some new indicators of residential satisfaction like Relief intervention, services from local govt. and others.

- Thanks for these valuable remarks. Taking the other two reviewers comments, we have amended the analytical approach's clarifications that may enrich the importance and methodological suitability of this paper. And we also believe that our findings also contribute to the methodological enhancement of residential satisfaction study. 

Since there is no formal institutional approval of the study it is important to understand how the security of the data is maintained. Further details about the both study areas regarding to the housing status, average monthly income and different govt. and other facilities will improve the paper. Also, the discussion on the policy implications can be further strengthened. Thorough editing of the paper is required to make it a standard paper for a reputed journal like PLOS ONE.

- There was no formal institutional review board approval for this empirical work, but this study was conducted under the master thesis project at the Economics Discipline of Khulna University, Bangladesh, and the field research methods were approved and tested, maintaining the cultural and social norms of the country.

- As per your suggestions, we have incorporated the background information in detail of the two communities (page no 4-7 and line no 94-116). In the conclusion section, the policy implication of this study also has been addressed in detail (page no 34 and line no 409-425). We have revised and edited the whole paper accordingly. 

Reviewer #2:

 • The author reported an important topic on going policy challenges; there are very few scientific evidence exist for researching on this challenging and highly sensitive issue by looking on transboundary migrations, social conflicts.

- Thanks for such valuable remarks; it motivates us for further research endeavours. 

The study objective is “how and to what extent the host communities' residential satisfaction level has been influenced due to Rohingya”. The word “Influence” needs to be revised. Looking on the results it has reported more on individual’s perception on residential satisfaction; but drawing an robust to Rohingya camp remain open (given that it often hard to make a clear boundary in presence of many other factors).

- The words influence has been revised and replaced with affected (page no 3 and line no 68).Thank you. 

- We agree with the remarks that our results represent the individual residential satisfaction perception of the community people living near the Rohingya camps; however, to see the spill-over effect, we consider two other communities in our sample far away from the Rohingya camps. So that our results also point out the satisfaction level who are not directly impacted by the community. 

• Include an operation definition of “neighboring”; in your sample case study unit they vary between 1 to 22 KM….

- Thanks for your suggestion. Accordingly, we have included the operational definition of the ‘neighbouring’. In this study, the neighbouring communities regard the people living around 2 kilometres of the Rohingya camp area. (page no 5 and line no 96-97).

• Mention some scopes of the study

- We have mentioned the scopes in detailed now (page no 3 line no 60-69). 

• Please highlight some criteria for survey unit (case study) selection – spatial and non-spatial; and mention about why the candidate unites were selected what limitation or sensitivity it may cause in the results. 

- We have edited and revised information about the study area selection criteria. We selected two types of villages to understand the residential satisfaction level at the neighbouring communities and compare them with the communities that are non-neighbouring. 

- We also focused on similar geographic and demographic characteristics to select the study sites. Please see the details in Section 2.1 (page no 4-7 and line no 94-116).

• The author included almost a comprehensive methodology; however, it might be helpful to link and discuss on Multicriteria approach as they are dealing with multivariate factors; therefore it is possible to find some cluster in the in the response variables and fine tune possibility of the model fit. See some example:

http://www.macrothink.org/journal/index.php/emsd/article/view/4874

https://www.sciencedirect.com/science/article/abs/pii/S2212420915301023

Thanks for your valuable remarks. Based on the existing literature, we have comprised all items into four clusters/components such as Satisfaction with Social Environment (SSE), Satisfaction with Neighbourhood Environment (SNE), Satisfaction with Public Service and Facilities (SPS&F) and Satisfaction with Dwelling Units (SDU). After that, we conducted the reliability (Alpha) test of each component. The Crohnbach’s alpha values of the components SSE, SNE, SPS&F and SDU are 0.906, 0.920, 0.828 and 0.847, respectively, indicating the convergent validity levels high and satisfactory, which are acceptable(page no 12 and line no 181-185). 

• In general, the analytical approach needs a revisit by highlighting the objective of the research. It has built some type of index on RS and tried to investigate to key factors.

- Thanks. We have revised the research objectives and reorganized the analysis accordingly (page no 3 and line no 67-72).

• In results, the fitting the linear regression curve is not much logical, where you the observation point are shows some clusters…it is obvious that RSI and SSE/SNE/RSI is correlated, the level might be highly due to the number of variables. Please check their sensitivity

- Thanks. Indeed, RSI and SSE/SNE/SPSF/SDU are correlated. Nevertheless, the correlation's intensity is not the same in SSE, SNE, SPSF, and SDU individually. In this regard, we performed linear regression curves to see the intensity of the components' relationship with the overall residential satisfaction and presented them graphically. However, we have conducted the Pearson correlation test, that it is not necessary to display it again. For this, we have deleted this from our manuscript. 

• The word “comparative” should be used with care. So far, the objective and results of the research is exploring and reporting the KI opinion or perceptions. It is hardly compare the situation after and before…where there is less firm-base to compare, rather than explore on the individual perception on current state.

- Here the variable “work opportunities” is categorical. We asked what was their level of working opportunities after the Rohingya Influx. Options included decrease, remain same and increased. Regarding reference categories “decrease,” we distinguished the satisfaction level of the people whose satisfaction level remained the same and increased. 

• The study has documented relevant literature, method and findings well; however still need some more re-look in discussion part. For instance, Need to generalize and link to theories and literature that has already been discussed earlier so that the scientific contribution become clear

- We have added new literature and also annotated our contributions beyond the existing state-of-the-art in discussion section (page no 34 and line 390-408). 

• The author may consider to include a conclusion by highlighting the key findings, limitations and way forward

- Thank. We have added a conclusion section by highlighting key findings, limitations, and the present future possible research-policy agenda (page no 34-35 and line no 409-425).

• There are some typo and minor errors which need to be revised as well

- We have revised the manuscript and addressed typos. Thank you.

• Considering all points raised, this could be one of the important contributions for multidisciplinary readers of Plos Journal - those are interested on the study regional and beyond.

good luck...

- Thanks for your valuable suggestions, and we hope this revised version has addressed all your requirements. 

Reviewer #3: 

This study aims to measure the level of residential satisfaction of the host community of Rohinga refugees, which is weakly addressed in this paper. Apart from the used tools to index the satisfaction level, the methodology is week. It would be necessary to know the level of satisfaction before the influx of the migrants so that the level of satisfaction can be compared.

- Thank you so much for your thoughtful comments. We have again revised the research objectives and rechecked the analytical parts and the result section accordingly.

- As we wish, you may agree with us that there was no base-line study on Residential satisfaction for the communities we studied and even though other parts of the country. The government collects information during the census, but hardly any of the censuses include any variable related to residential satisfaction measurement. There is no such kind of practice in the country. Moreover, even we are for sure, if there were no Rohingya influx, we (all the researchers/ development workers) would not put our attention to those communities for hardly any kind of research. So we have to believe in the perception of wise comparison at the individual level, and there are significant studies in social sciences which conduct the same approaches to understand the past and present, for example, studies on disaster risk research, livelihood quality assessment etc. However, we have revised our methodology section carefully to remove this kind of confusion; please check the details in section 2.1. Thank for your understanding. 

1. The sampling technique is not straightforward- it is just not enough to say that the HH was randomly selected. It is okay even if the sample is not randomly selected if backed by valid motivation. In sampling, there is no indication of, discussion of the Non-Ukhiya group. How do we know if Ukhiya and Non-Ukhiya groups are comparable? We have to know if these groups are comparable by education, income, and other basic household characteristics. [Solution]

- Thanks for your valuable observations. We have revised the random selection process. Besides, we have now renamed the Ukhiya communities and Non-Ukhiya communities as ‘neighbouring’ and ‘non-neighbouring” community, respectively, and their operational definition is presented in the methodology section (page no 5 and line no 96-97).

- Again, both of these communities have almost the same geographic and demographic characteristics. However, from these two communities, the household heads were randomly interviewed form four or five houses next (page no 7 line no 121-123). 

2. Figure 7 compared the satisfaction level of Non-Ukhiya with the Ukhiya community and tried to establish that the cause of the difference is due to the influx of migrants (line 269). This is a very weak approach to establish this causality because we do not know the level of satisfaction of Ukhiya and Non-Ukhiya community before the influx of migrants. We only see the difference after the influx. The direction of difference can go to many ways. The difference may exist even before the influx of the migrants (outcomes of table 4 almost confirms the difference is pre-influx).

- As mentioned above, in our two study sites, one is nearer to the Rohingya camp areas (within 2 km); the other is distant from the camp area where there is no Rohingya presence. These study areas are located in the same geographical zone, and the socio-demographic features are the same. Please see section 2.1 for details. As there is no evidence or record of residential satisfaction before the influx of Rohingya, this study intended to predict the possible residential satisfaction level. We agree that before the Rohingya influx, there must be a variation of residential satisfaction and the factors; therefore, the measurement factors/influential factors could be different. In this study, we took an influx of Rohingya is one of the significant contributors to residential satisfaction, and it was so examined in our study. 

3. Why are 5 different models used to evaluate the level of satisfaction is not clear (precisely- each model is OLS with a different combination of socio-demographic variables)? What is the motivation of each model?

- Here, we have run the regression model five times to explore the effect of every single component of residential satisfaction and reduce the severity of multicollinearity. Thank you. 

4. L155- I wonder if authors used a Likert scale, why do they need RSI ? There is a need for a detailed literature review backing this index's use.

- Literature shows that the most common approach of assessing the residential satisfaction is the self-evaluation either by asking the general question regarding the degree of satisfaction with residents’ environment or asking the level of satisfaction with different aspects or components that, in some form, results in residential satisfaction index. However, in this regards, we based on the literature, a residential satisfaction index has been developed with twenty-two satisfaction aspects that are grouped with four residential satisfaction components. Thank you. 

5. As this study is about HH satisfaction analysis, OLS may not be the better approach, I would recommend authors to play with other commonly used satisfaction analysis methods such as Factors Analysis and Principal Components Analysis. These will enable identifying laten variables asoociated with satisfaction and preventing multicollinearity effects n the model estimations.

- "We beg to differ on this suggestion by the reviewer. It is because the suggested approach is a multivariate approach for which concerned data need to be highly correlated. In line with our study objective and research questions (where we did not intend to discover any latent variable) along with the nature of the collected data, a multivariate approach of analysis (i.e., PCA) is unlikely to be suitable for our dataset. Therefore, we would like to proceed with the regression results presented in Table 6.

---

## [Editor Report · Decision Letter 1]

15 Apr 2021

Analysis of residential satisfaction: An empirical evidence from neighbouring communities of Rohingya camps in Cox's Bazar, Bangladesh

PONE-D-21-02960R1

Dear Authors,

We’re pleased to inform you that your manuscript has been judged scientifically suitable for publication and will be formally accepted for publication once it meets all outstanding technical requirements.

Kind regards,

Professor Hafiz T.A. Khan, Ph.D, CStat

Academic Editor

PLOS ONE
---

## [Editor Report · Acceptance letter]

19 Apr 2021

PONE-D-21-02960R1 

Analysis of residential satisfaction: An empirical evidence from neighbouring communities of Rohingya camps in Cox's Bazar, Bangladesh 

Dear Dr. Biswas:

I'm pleased to inform you that your manuscript has been deemed suitable for publication in PLOS ONE. Congratulations! Your manuscript is now with our production department. 

Kind regards, 

on behalf of

Professor Hafiz T.A. Khan 

Academic Editor

PLOS ONE